# Seasonal effects of photophysiology and chlorophyll *a* abundance on phytoplankton group-specific primary production in the Kuroshio region as revealed by SeaStar/SeaWiFS

### 5 Takafumi Hirata<sup>1</sup>, Koji Suzuki<sup>1</sup>

<sup>1</sup>Faculty of Environmental Earth Science, Hokkaido University, Sapporo, 060-0810, Japan *Correspondence to*: Takafumi Hirata (tahi@ees.hokudai.ac.jp)

### Abstract.

To evaluate the group-specific primary production of diatoms, haptophytes, and cyanobacteria in the

- Kuroshio region, a novel satellite observation methodology using the SeaStar/SeaWiFS satellite instrument was developed. The method used bio-optical relationships between the group-specific production and bio-optical properties such as the photosynthetic quantum yield and chlorophyll *a* specific optical absorption coefficient of phytoplankton, the last two of which were also estimated together with the group-specific production rather than assumed a priori. A global property of the
- absorption coefficient of phytoplankton, that the coefficient value at the wavelength of 510 nm was close to their spectral average, was highlighted in the method for a use of multi-spectral ocean color satellite data. Our results showed that the derived quantum yield index was higher for diatoms than haptophytes and cyanobacteria. Furthermore, intraspecific variation in the index, emerged as a latitudinal gradient: the values for cyanobacteria increased towards the higher latitudes. The group-
- specific primary production in the Kuroshio region showed that the climatological average of 134, 72 and 40 mg C m<sup>-2</sup> day<sup>-1</sup> for diatoms, haptophytes, and cyanobacteria, respectively. A comparison among variability of the group-specific primary production, the quantum yield index, and the absorption coefficient suggested that, in the Kuroshio region, the primary production due to diatoms was driven by their abundance through the year, whereas that due to cyanobacteria by photophysiology. The
- production due to haptophytes was seasonally regulated by both abundance and photophysiology.

### **1** Introduction

The Kuroshio Current is the western boundary current carrying warm oligotrophic waters from the subtropic to the sub-arctic in the southwest region of the North Pacific (Fig. 1). Despite the oligotrophy of the current, higher fishery production has been recognized in and around the current, which is a paradox of the Kuroshio ecosystems. Understanding the physical-biogeochemical-biological

paradox of the Kuroshio ecosystems. Understanding the physical-biogeochemical-biological mechanisms of the Kuroshio ecosystems is crucial to not only explain this paradox as a science, but also provide a scientific basis for so-called ecosystem-based fishery management (Pikitch et al., 2004).

Recent advances in fishery sciences have revealed that the Kuroshio Current provides habitats for the larvae of pelagic fish as well as spawning grounds upstream (e.g., Sassa et al., 2004). In

- addition, the oligotrophic Kuroshio Current meets a cold and nutrient-rich subarctic current, the Oyashio Current, forming a heterogeneous sub-polar front and eddies where primary production is fueled by vertical nutrient fluxes associated with physical instabilities at the front via upward transport of nutrients (Nagai et al., 2012; Clayton et al., 2014). In such a turbulent environment, diatoms with higher nutrient and/or low grazing mortality rates can become predominant in the phytoplankton
- assemblages. Because diatoms have a dense silica wall and sink well in calm water, it is clear that turbulent upward motion is needed to keep them in the water column and is a prerequisite for the formation of dense populations of diatoms (Mann and Lazier, 2006). It is well known that many great fisheries are dependent on a grazing food chain that starts at diatoms and proceeds via copepods to young fish (Cushing, 1989). Therefore, understanding the community structure of primary producers 45 provides clues of the underlying ecological structure.

In a case study in the North Atlantic, Claustre et al. (2005) derived primary production parameters by the multiple regression analysis of in situ data and demonstrated that they were dependent on the phytoplankton community structure, and Uitz et al., (2008) further parameterized them. However, the Kuroshio region was not included in their analysis. Remote estimations of size-

50 fractionated primary production have also been conducted (Kameda and Ishizaka, 2005; Uitz et al., 2010) covering larger geographical extents to support intermittent efforts of in situ observations of size-fractionated or taxon-specific primary production (Owens et al., 1993; Jochem et al., 1995; Lee et al., 1996; Bury et al., 2001; Claustre et al., 2005; Moran et al., 2004; Barnes et al., 2014; Robinson et al., 2017). While remote sensing of size-specific primary production revealed a synoptic view of

physiological properties in association with the ambient environment such as the sea temperature (Carr et al., 2006).

Our objective in this study is to estimate the regional, geographical patterns of the groupspecific primary production (i.e., diatoms, haptophytes, and cyanobacteria) and the related 60 photosynthetic properties in the Kuroshio Current and its adjacent waters in the North Pacific. We have developed a novel remote sensing methodology by integrating an image processing technique with a bio-optical model, and relaxing the assumptions of a certain physiology-environment association made in the previous approaches: the group-specific photophysiological properties, such as the quantum yield index (i.e. a scaled quantum yield) and the chlorophyll-specific absorption coefficient, are also obtained from space in this paper.

05 Obtained from space in this paper

### 2 Material and Method

### 2.1 Material

### 2.1.1 In situ data

- In this analysis, we defined the Kuroshio region as the area between 120 to 160°E and 24 to 44°N, which includes the Kuroshio Current itself (shown schematically as a white arrow in Fig. 1) and adjacent waters. The southern side of the Kuroshio is characterized as a subtropical water whereas the northern side as a sub-polar water. We used in situ measurements of phytoplankton pigments measured with High Performance Liquid Chromatography (HPLC) and the spectral optical absorption coefficient
- of phytoplankton,  $a_{ph}(\lambda)$  where a wavelength is denoted by  $\lambda$ . The HPLC pigments were used to calibrate a satellite algorithm, OC-PFT (Hirata et al., 2011), to derive the phytoplankton composition in the Kuroshio region, while the absorption coefficient was used in a bio-optical model to estimate the group-specific primary production.

Water samples for the HPLC pigment analysis (Suzuki et al., 2015) were collected during
several cruises: the R/V *Tansei-Maru* KT-12-31 cruise from November 18 to 22, 2012 (Fig. 1).
Diagnostic Pigment Analysis (Vidussi et al., 2001; Uitz et al., 2006) was then applied to define the pigment-based taxonomic composition of the phytoplankton. Only the surface data (< 10 m) was used in our analysis (N = 28).</li>

- The absorption data was only available for the R/V *Soyo-Maru* SY-13-04 cruise. To increase robustness of our analysis, all absorption data from the surface to the maximum of 59 m were included in our analysis (n = 17). As shown in Section 3.1, that did not affect our result but rather supported the general applicability of our bio-optical model over different spaces (horizontally and vertically). The absorption coefficient was measured using the filter pad technique (Mitchell et al., 2000) with a pathlength correction scheme (Cleveland and Weidemann, 1993). We also used surface absorption data
- from geographically distinct regions, such as the Benguela upwelling waters (Fishwick et al., 2006) and other globally located regions (Werdell and Bailey, 2005), to support robustness of a bio-optical model.

### 2.1.2 Satellite data

- Our estimation of the group-specific primary productivity requires satellite data of (1) the 95 chlorophyll *a* concentration, *Chla* [mg m<sup>-3</sup>]; (2) the absorption coefficient of the total phytoplankton community at 510 nm,  $a_{ph}$  [m<sup>-1</sup>]; (3) the primary production of the total phytoplankton community, *PP* [mg C m<sup>-2</sup> day<sup>-1</sup>]; and (4) the Photosynthetically Available Radiation, *PAR* [µmol m<sup>-2</sup> day<sup>-1</sup>] as inputs. The Level 3 SeaWiFS (Sea-viewing Wide Field of View Sensor) monthly 9 km remote sensing reflectance,  $R_{rs}$  [sr<sup>-1</sup>], at wavelengths of 412 nm, 443 nm, 490 nm, 510 nm, and 555 nm, *Chla*, and PAR
- for the period of 1998–2007 were obtained from NASA. The remote sensing reflectance was inverted to derive  $a_{ph}$  using the inversion model of Smyth et al. (2006). In addition, *Chla* derived from  $R_{rs}$ (O'Reilly et al., 1998) was used to derive the relative abundance of the three phytoplankton taxonomic groups (i.e., diatoms, haptophytes, and cyanobacteria [%]) following the procedure of Hirata et al. (2011), which was specifically calibrated for in situ data of the phytoplankton community structure in
- the Kuroshio region described in the previous section; see Fig. 2. The relative abundance of each phytoplankton group was subsequently multiplied by Chla to obtain the group-specific Chla, that is, Chla<sub>i</sub> [mg m<sup>-3</sup>]. Values of PP for the total phytoplankton community derived from the Carbon-based Production Model (CbPM, Westeberry et al., 2008) with the same temporal and spatial resolutions as SeaWiFS products Oregon University the were obtained from State 110 (http://www.science.oregonstate.edu/ocean.productivity/).

### 2.2 Methodology

### 2.2.1 Bio-optical model of taxon-specific primary production

Because most of the satellite signal originates from the surface of a water column, the depth dependency of the quantities described hereafter will be omitted for brevity, unless otherwise stated.

Primary productivity of the total phytoplankton community is expressed by the quantum yield of 120 photosynthesis,  $\Phi$  [mg C µmol<sup>-1</sup>],  $a_{ph}$ , and PAR:

$$PP = \Phi \cdot \int_{400}^{700} E_0(\lambda) \cdot a_{ph}(\lambda) d\lambda$$
  
=  $\Phi \cdot PAR^{eff} \cdot \int_{400}^{700} a_{ph}(\lambda) d\lambda$   
=  $300 \cdot \Phi \cdot E^* \cdot PAR \cdot \chi_{aph}(\lambda) \cdot a_{ph}(\lambda)$  (1)

- where  $\lambda$  is the wavelength [nm] and  $E_0$  is the quantum number [ $\mu$ mol m<sup>-2</sup> day<sup>-1</sup>], which is related to PAR via  $PAR = \int_{400}^{700} E_0(\lambda) d\lambda$ . The effective PAR (PAR<sup>eff</sup>) [ $\mu$ mol m<sup>-2</sup> day<sup>-1</sup>] is defined by  $\int_{400}^{700} E_0(\lambda) a_{ph}(\lambda) d\lambda / \int_{400}^{700} a_{ph}(\lambda) d\lambda$ , and  $E^*$  [dimensionless] is the ratio between PAR<sup>eff</sup> and the actual PAR. Finally,  $\chi_{aph}(\lambda)$  is a factor converting  $a_{ph}(/)$  to its spectral average,  $\overline{a_{ph}(\lambda)}$ , over the range of 400–700 nm, that is,  $\chi_{aph}(\lambda) = (\int_{400}^{700} a_{ph}(\lambda) d\lambda/300) / a_{ph}(\lambda) = \overline{a_{ph}(\lambda)} / a_{ph}(\lambda)$ . Equation 130 (1) shows that *PP* is proportional to  $\chi_{aph}(\lambda) \cdot a_{ph}(\lambda) (= \overline{a_{ph}})$  with a light-dependent proportional factor,  $300 \cdot \Phi \cdot E^* \cdot PAR$ . Previous in situ observations (Marra et al., 2007) demonstrated a linear
- relationship between *PP* and  $\overline{a_{ph}(\lambda)}$ , with different slopes depending on the study site. Thus, Eq. (1) explains the observation. On the other hand, a care must be taken that the proportionality factor,  $300 \cdot \Phi \cdot E^* \cdot PAR$  is site-dependent, and can vary temporally and spatially, so that a comparison
- between *PP* and  $\chi_{aph}(\lambda) \cdot a_{ph}(\lambda) (= \overline{a_{ph}(\lambda)})$  can also lead to non-linearity if the data pairs of *PP* and  $\chi_{aph}(\lambda) \cdot a_{ph}(\lambda)$  (or  $\overline{a_{ph}(\lambda)}$ ) are obtained at different locations (both horizontally and vertically) and/or time (Hirata et al., 2009).

Our data from the Kuroshio region show a hinge point at approximately 510 nm in the  $\chi_{aph}(\lambda)$  spectra (Fig. 3a). It corresponds to the central wavelength of one of the SeaWiFS wavebands

- but neither of MODIS (Moderate Resolution Imaging Spectroradiometer) onboard the Aqua/Terra satellites nor VIIRS (Visible Infrared Imaging Radiometer Suite) onboard the Suomi National Polar-Orbiting Partnership satellite. Values of  $\chi_{aph}(510)$  are approximately 0.973, close to unity. The coefficient of variation (CV) at 510 nm also displays one of the lowest values, 0.035 (Fig. 3). The same analysis for the biologically productive Benguela upwelling waters, which are geographically and
- ecologically different from the Kuroshio region, shows  $\chi_{aph}(510) = 0.882$  (Fig. 3b). The CV (0.035) at 510 nm is close to the minimum (0.034) found at 501 nm (the wavelength of 501 nm is however not used by historical ocean colour instruments). Furthermore, the global dataset (the NASA bio-Optical Marine Algorithm Dataset, NOMAD, Werdell and Bailey 2005) returned  $\chi_{aph}(510) = 0.992$  with one of the lowest CV values of 0.087, when discrete measurements of  $a_{ph}(\lambda)$  is interpolated over the
- spectrum (Fig. 3c). The small variability in χ<sub>aph</sub>(510)(= 0.949 ± 0.06) over the three different datasets shows the quasi-constant characteristic of χ<sub>aph</sub>(510) despite the large geographical differences between these datasets. Subsequently, a<sub>ph</sub>(λ) can often be approximated by χ<sub>aph</sub>(510) · a<sub>ph</sub>(510) worldwide with a quasi-constant value of χ<sub>aph</sub>(510). Note that our data from the Kuroshio included not only horizontal but also vertical distribution of a<sub>ph</sub>(λ), supporting this approximation. The small variability in χ<sub>aph</sub>(510) over large geographical and temporal scales implicitly indicates that
- small variability in  $\chi_{aph}(510)$  over large geographical and temporal scales implicitly indicates that  $\chi_{aph}(510)$  is also quasi-constant irrespective of phytoplankton community structure.

When Eq. (1) is applied to each phytoplankton group, the group-specific primary production  $PP_i$  is expressed by

$$PP_i = A_i^{510} \Phi_i^{\#}$$
 (2)

where the quantum yield index is given by  $\Phi_i^{\#} = \Phi_i \cdot E_i^*$  and the absorbed PAR by  $A_i^{510} = 300 \cdot PAR \cdot \chi_{aph}(510) \cdot a_{ph,i}(510)$ . Meanwhile, the primary productivity of the total phytoplankton community *PP* is a linear sum of those of each taxonomic group  $PP_i$  (i.e.  $PP = \sum_{i=1}^{Np} PP_i$ ), where *i* is the index for

phytoplankton taxonomic groups and Np is total number of phytoplankton groups. When diatoms are represented by i=1, haptophytes by i=2, cyanobacteria by i=3, and any other phytoplankton groups by i=4, respectively (i.e.  $N_p=4$ ). N measurements of total PP allow us to establish N number of  $PP = \sum_{i=1}^{Np} PP_i$  such that:

(3)

$$PP = A\Phi_i^{\#} + \epsilon_{pp}$$

where the vector  $\boldsymbol{PP} = [PP_{n=1}, \dots, PP_N]^T$  consists of N elements (measurements) of total PP, and the vector  $\boldsymbol{\Phi}_i^{\#} = [\boldsymbol{\Phi}_{i=1}^{\#}, \dots, \boldsymbol{\Phi}_{i=3}^{\#}]^T$  has three elements consisting of a  $\boldsymbol{\Phi}_i^{\#}$  value for diatoms, haptophytes and cyanobacteria, respectively. The vector  $\boldsymbol{\epsilon}_{pp} = [PP_{i=4,n=1}, \dots, PP_{i=4,n=N}]^T$  contains N elements consisting of PP of any phytoplankton groups (i=4) other than those mentioned above.

*N* elements, consisting of *PP* of any phytoplankton groups (*i*=4) other than those mentioned above.
Finally, the matrix 
$$\mathbf{A} = \begin{bmatrix} A_{i=1,n=1}^{510} & \cdots & A_{i=3,n=1}^{510} \\ \vdots & \ddots & \vdots \\ A_{i=1,n=N}^{510} & \cdots & A_{i=3,n=N}^{510} \end{bmatrix}$$
 has *N* x 3 elements consisting of *N* number of  $A_i^{510}$ 

for the three taxonomic groups (diatoms, haptophytes and cyanobacteria). When *PP* and *A* are known, Eq. (3) can be solved to obtain  $\Phi_i^{\#}$  as the least square solution  $\Phi_i^{\#} = (A^T A)^{-1} A^T PP$ , where  $(A^T A)^{-1}$ is the inverse of the matrix  $A^T A$ . While the primary production of total phytoplankton community can

- be obtained using satellite data (Behrenfeld et al., 2005; Westberry et al., 2008), determination of the matrix of the absorbed PAR *A* requires χ<sub>aph</sub>(510), PAR and a<sub>ph,i</sub>(510). We use the value 0.949 for χ<sub>aph</sub>(510) in the virtue of its quasi-constant characteristic. Satellite data of PAR is also available (e.g. Frouin et al., 2012). Hence, only a<sub>ph,i</sub>(510) needs to be known. Sections 2.2.2 explains how to obtain a<sub>ph,i</sub>(510). Once a<sub>ph,i</sub>(510) is derived, *A* will be known. Using *A* and *PP*, Φ<sub>i</sub><sup>#</sup> can be derived with Eq. (3), with which the group-specific primary production for diatoms, haptophytes and cyanobacteria
- Eq. (3), with which the group-specific primary production for diatoms, haptophytes and cyanob is finally obtained by Eq. (2).

Hereafter, the quantum yield index  $\Phi_i^{\#}$  will be invoked to represent  $\Phi_i$  or the state of the photophysiology of each phytoplankton group, even though  $\Phi_i^{\#} (= \Phi_i \cdot E_i^*)$  is not precisely the same quantities as  $\Phi_i$ . We will also invoke  $a_{ph,i}(510)$  to represent the group-specific *Chla* biomass due to the tight correlation between  $a_{ph}(510)$  and *Chla* found in the local and global data ( $r^2 = 0.90$ , p <

the tight correlation between  $a_{ph}(510)$  and *Chla* found in the local and global data (r<sup>2</sup> = 0.90, p

$$a_{ph}(510) = \sum_{i} \{a_{ph,i}^{*}(510) \cdot Chla_{i}\},\tag{4}$$

where  $a_{ph,i}^*(510)$  is the chlorophyll *a* specific absorption coefficient for each phytoplankton group defined as  $a_{ph,i}^*(510)=a_{ph,i}(510)/Chla_i$ . In the manner similar to formulating Eq. (3), *N* measurements of  $a_{ph}(510)$  allows us to establish *N* number of Eq. (4) such that

$$a_{ph}^{510} = Chla_i a_{ph,i}^{*510} + \epsilon_{aph}$$
<sup>(5)</sup>

where  $a_{ph}^{510} = [a_{ph}(510)_{n=1} \dots a_{ph}(510)_N]^T$  consists of N elements of  $a_{ph}(510)$ ,  $a_{ph,i}^{*510} = [a_{ph,i=1}^{*510}, \dots, a_{ph,i=3}^{*510}]^T$  is a 3-element vector consisting of  $a_{ph,i}^*(510)$  for daiatoms, haptophytes and cyanobacteria,  $\epsilon_{aph} = [a_{ph,i=4}(510)_{n=1}, \dots, a_{ph,i=4}(510)_{n=N}]^T$  consists of N elements of  $a_{ph,i}(510)$  of any other phytoplankton groups  $(a_{ph,4}(510))$  and the matrix 210  $Chla_i = \begin{bmatrix} Chla_{i=1,n=1} & \cdots & Chla_{i=3,n=1} \\ \vdots & \ddots & \vdots \\ Chla_{i=1,n=N} & \cdots & Chla_{i=3,n=N} \end{bmatrix}$  has N x 3 elements consisting of N number of  $Chla_i$  for

diatoms, haptophytes and cyanobacteria (i.e. N x 3 elements). Equation (5) can be solved to obtain  $a_{ph,i}^{*510}$  as the least square solution  $a_{ph,i}^{*510} = (Chla_i^T Chla_i)^{-1} Chla_i^T a_{ph}^{510}$ , where  $(Chla_i^T Chla_i)^{-1}$  is the inverse of the matrix  $Chla_i^T Chla_i$ . The solution requires  $a_{ph}(510)$  and  $Chla_i$  to be known. Both can be obtained as satellite data (e.g., Lee et al., 2002, Smyth et al., 2006, Werdell et al., 2013, Bracher

et al., 2009, Hirata et al., 2011, Sadeghi et al., 2012). As a result, a<sup>\*510</sup><sub>ph,i</sub> can be obtained. Once a<sup>\*510</sup><sub>ph,i</sub> is derived, we can find a<sub>ph,i</sub>(510) for diatoms, haptophytes and cyanobacteria as an element of the vector *Chla<sub>i</sub>* a<sup>\*510</sup><sub>ph,i</sub>. The resulting a<sub>ph,i</sub>(510) is then used to obtain the matrix *A* for determining Φ<sup>#</sup><sub>i</sub> by Eq. (3), hence determining *PP<sub>i</sub>* eventually.

### 220 2.2.3 Spatial data sub-sampling

Eqs. (3) and (5) cannot be preformed for each grid or pixel in a satellite image because only one value is available at each pixel for each input variable (e.g., only one value of  $Chla_i$  is available for (x, y),

where x and y represent the longitudinal and latitudinal coordinates, respectively). However, they are solvable when N number of neighboring pixels are used. By defining a geographically small region (or "window") consisting of  $N=n \times n$  neighboring pixels in a satellite image of each input variable (i.e.  $N=n^2$  satellite data samples are contained in the window), we can sub-sample the satellite data from the window to collect N measurements. We selected n=5 (i.e.  $N = 5 \times 5 = 25$  pixels) for a square window in the present analysis in order to achieve a balance between the statistical robustness of the regression

- and the resulting degradation of the spatial resolution that was tolerable for our later analysis. With n=5, the system of the simultaneous equations (i.e. Eq. (5)) established for the window is usually overdetermined as there are only three unknowns (i.e.  $a_{ph,i}^*(510)$  for the three phytoplankton groups). Hence, the system is solved by the least square method. One value of  $a_{ph,i}^*(510)$  of each phytoplankton group (3 groups in our case) is then obtained for that window. By repeating this operation for
- neighboring windows within the same satellite image, one can obtain a map of  $a_{ph,i,i}^{*}(510)$  as a collection of output values from these windows, although the output images from this procedure has less spatial resolution than those of the input variables. Once  $a_{ph,i,i}^{*}(510)$  is derived,  $a_{ph,i}(510)$  is obtained, too, by multiplying  $a_{ph,i}^{"}(510)$  by *Chla<sub>i</sub>*. Using  $a_{ph,i}(510)$ , the similar procedure is repeated with Eq. (3), which then gives a map of  $\Phi_{i}^{\#}$ , hence of *PP<sub>i</sub>*, for the 3 groups.
- We further repeat these operations for monthly images to generate a monthly time series of the derived variables (a<sup>\*</sup><sub>ph,i</sub>(510), a<sub>ph,i</sub>(510), Φ<sup>#</sup><sub>i</sub>, and PP<sub>i</sub>) for the period of 1998- 2007, from which monthly climatological data is obtained. Note that, for each window, a correction of the degree of freedom may be required in the regression analysis when a significant spatial autocorrelation among data samples within a window of an input variable(s) is found. Also note that the degree of tolerable degradation of spatial resolution of derived quantities is application-specific, and one may change the
  - window size as appropriate.

As a result, the integration of the bio-optical theory and the multi-pixel image processing enables the derivation of the biological quantities  $a_{ph,i}^*(510)$ ,  $a_{ph,i}(510)$ ,  $\Phi_i^{\#}$ , and  $PP_i$  for each phytoplankton group considered here.

**2.2.4 Evaluation of the relative impact of**  $a_{ph,i}(510)$  and  $\Phi_i^{\#}$  on  $PP_i$ 

Multiple linear regression analysis was performed between the dependent variable  $PP_i$  and the independent variables  $a_{ph,i}(510)$  and  $\Phi_i^{\#}$  using their monthly climatology over 1997–2007. All variables were standardized (i.e. a mean value was subtracted from the original data, and a resultant value was divided by the standard deviation) prior to the regression analysis. The analysis was performed using spatial data within the Kuroshio region for each month. The multiple linear regression coefficients for the standardized  $a_{ph,i}(510)$  and  $\Phi_i^{\#}$  were defined as their contributions to  $PP_i$ , and seasonal variation of the contributions was evaluated for each taxonomic group.

### **3 Results**

## **3.1** Spatial distribution of the group-specific characteristics associated with primary production in the Kuroshio region

Figure 4 shows the climatological distribution of the absolute abundance of  $Chla_i$ , the relative abundance of  $Chla_i$ ,  $a_{ph,i}(510)$ ,  $a_{ph,i}^*(510)$ ,  $\Phi_i^{\#}$ , and  $PP_i$  of the three taxonomic groups for the period of 1998–2007. The climatological average of the absolute abundance of  $Chla_i$  of diatoms, haptophytes, and cyanobacteria of the entire region defined in our analysis were 0.21 mg Chla m<sup>-3</sup>, 0.11 mg Chla m<sup>-3</sup>

- <sup>3</sup>, and 0.03 mg Chla m<sup>-3</sup>, respectively. While the spatial average of  $Chla_i$  depends on the region defined, we found the clear tendency that the diatom- and haptophyte-derived  $Chla_i$  were higher on the northern side of the Kuroshio Current. Conversely, cyanobacterium-derived  $Chla_i$  were rather uniform over the region.
- The spatial pattern of  $a_{ph,i}(510)$  was similar to that of  $Chla_i$  for all three phytoplankton 275 groups (Figs. 4a-c and 4g-i); therefore,  $a_{ph,i}(510)$  can be used to represent the  $Chla_i$  in the Kuroshio region even if 510nm is out of chlorophyll a absorption bands. The climatological 10-year averages (1998–2007) of  $a_{ph,i}(510)$  for diatoms, haptophytes, and cyanobacteria were 0.0016 m<sup>-1</sup>, 0.0026 m<sup>-1</sup>, and 0.0012 m<sup>-1</sup>, respectively, indicating that haptophytes are more abundant over this region.
- The relative abundance of diatoms also showed a similar spatial pattern to *Chla<sub>i</sub>*. However, haptophytes and cyanobacteria did not. Haptophytes exhibited a higher relative abundance along the Kuroshio Current (Fig. 4e), while cyanobacteria did so on the southern side of the Kuroshio Current (Fig. 4f) (see Fig. 1 also). The climatological averages of the relative abundances of diatoms,

haptophytes, and cyanobacteria over the region were 18%, 30%, and 17%, respectively, indicating again that haptophytes are the most dominant group in the region

- The climatological regional averages of a<sup>\*</sup><sub>ph,i</sub>(510) for diatoms, haptophytes, and cyanobacteria were 0.016 m<sup>2</sup> mg Chla<sup>-1</sup>, 0.027 m<sup>2</sup> mg Chla<sup>-1</sup>, and 0.040 m<sup>2</sup> mg Chla<sup>-1</sup>, respectively. The magnitude of a<sup>\*</sup><sub>ph,i</sub>(510) for diatoms was smaller than that for cyanobacteria over the entire region, while the magnitude of a<sup>\*</sup><sub>ph,i</sub>(510) for haptophytes fell between those two values. The geographical variability in a<sup>\*</sup><sub>ph,i</sub>(510), defined here as the coefficient of variation, σ<sub>aph\*,i</sub>(510)/290 a<sup>\*</sup><sub>ph</sub>(510), where σ<sub>aph\*,i</sub>(510) is the standard deviation of a<sup>\*</sup><sub>ph,i</sub>(510), was higher for cyanobacteria
- than diatoms in the climatological field (0.10, 0.15, and 0.16 for diatoms, haptophytes, and cyanobacteria, respectively). An elevated value of  $a_{ph,i}^*(510)$  (> 0.05) was observed along the Kuroshio Current for cyanobacteria (Fig. 4i). Haptophytes showed a spatial pattern of  $a_{ph,i}^*(510)$ similar to that of *Chla<sub>i</sub>*, whereas diatoms showed a relatively uniform spatial pattern of  $a_{ph,i}^*(510)$ 295 than those of haptophytes and cyanobacteria.

The spatial distribution of  $\Phi_i^{\#}$  was classified into two patterns for the three phytoplankton groups. The diatom-derived  $\Phi_i^{\#}$ , with a climatological regional average of  $0.8 \times 10^{-3}$ , was higher on the southern side of the Kuroshio Current ( $\sim 0.2 \times 10^{-2}$ ) than on the northern side ( $< 0.1 \times 10^{-2}$ ). The haptophyte-derived  $\Phi_i^{\#}$  had a similar pattern to that of the diatoms but was relatively more uniform over the region with a climatological regional average of  $0.2 \times 10^{-3}$ . Conversely, the cyanobacteriumderived  $\Phi_i^{\#}$  (climatological regional average of  $0.2 \times 10^{-3}$ ) was to some extent higher on the northern side ( $\sim 0.5 \times 10^{-3}$ ) of the Kuroshio Current than on the southern side ( $\sim 0.2 \times 10^{-3}$ ), contrasting with the spatial patterns of the diatom- and haptophyte-derived  $\Phi_i^{\#}$ .

- The climatological regional averages of  $PP_i$  for diatoms, haptophytes, and cyanobacteria were 305 134 mg C m<sup>-2</sup> day<sup>-1</sup>, 72 mg C m<sup>-2</sup> day<sup>-1</sup>, and 40 mg C m<sup>-2</sup> day<sup>-1</sup>, respectively. The values of  $PP_i$ showed a clear latitudinal gradient for all groups such that they were higher at higher latitudes and lower at lower latitudes. However, the gradient was largest for diatoms and smallest for cyanobacteria. The latitudinal gradient of  $PP_i$  was therefore in agreement with  $Chla_i$  for diatoms but not necessarily so for cyanobacteria. As a result, the diatom-derived  $PP_i$  had a spatial pattern very different from that of
- the diatom-derived  $\Phi_i^{\#}$ , whereas the cyanobacterium-derived *PP<sub>i</sub>* had a pattern closer to that of the cyanobacterium-derived  $\Phi_i^{\#}$ .

## **3.2** Factors controlling group-specific primary production: which is more important, chlorophyll *a* biomass or physiology?

Figure 5 shows the relative contribution of variability in the group-specific phytoplankton abundance represented by  $a_{ph,i}(510)$  and the photophysiology represented by  $\Phi_i^{\#}$  to the variability in  $PP_i$ . The contribution of the abundance and photophysiology were not equal in general and depended on the taxonomic group and the season. For diatoms, the contribution of the abundance to  $PP_i$  was always

- larger than that of the photophysiology. The abundance contribution for diatom  $PP_i$  was larger than the physiological contribution throughout the year and was particularly large between May and November (0.77–0.85) compared to between December and April (0.38–0.64). Conversely, the photophysiological contribution to  $PP_i$  if diatoms were small or even absent over the all seasons (0–0.24). In general, diatom  $PP_i$  was derived by the abundance. For haptophytes, the physiological contribution remained
- large throughout the year, with a relatively smaller contribution (~0.58) in boreal summer (June– September) and a relatively larger contribution (~0.89) in winter (November–April). Conversely, the abundance contribution was highest (0.65) in June and lowest (0.25) in January. Thus, the two contributions were out of phase for haptophytes; however, this does not indicate a simple alternation between the abundance and physiological contributions over the year. Both the abundance and
- physiological contributions were found significant from May to November, whereas the physiological contribution dominated from December to April. For cyanobacteria, the physiological contribution always prevailed throughout the year (> 0.65), while a biomass contribution was also found, but only to a small degree (< 0.26).

#### 335 4 Discussion

The highest climatological regional average of primary production was found for diatoms in the entire Kuroshio region (134 mg C m<sup>-2</sup> day<sup>-1</sup>, 72 mg C m<sup>-2</sup> day<sup>-1</sup>, and 40 mg C m<sup>-2</sup> day<sup>-1</sup> for diatoms, haptophytes, and cyanobacteria, respectively). However, within the Kuroshio Current and its extension

domain (the area between two dotted curves in Fig. 4p and 4q), the  $PP_i$  for haptophytes was found to be higher than that for diatoms. Even though a direct validation of the satellite estimate of  $PP_i$  remains to be conducted, a recent in situ observation by Nishibe et al. (2015) within the Kuroshio domain showed that smaller phytoplankton (< 10 µm) have higher production (61–185 mg C m<sup>-2</sup> day<sup>-1</sup>) than larger