# Peer review of "Seasonal effects of photophysiology and chlorophyll *a* abundance on phytoplankton group-specific primary production in the Kuroshio region as revealed by SeaStar/SeaWiFS"

_Biogeosciences, 2017_

## Referee Comment (RC1) · Anonymous Referee #1 · 27 Jun 2017

Because different phytoplankton has different response to light and nutrients, it is always desired to know group (or class or species) specific photosynthesis rate, as such information can also improve the estimation of global primary production. To meet this desire and demand, the authors developed specific primary production in the Kuroshio region for three "functional groups" of phytoplankton: diatoms, haptophytes, and cyanobacteria. Subsequently, the authors described their spatial/temporal variations as well as the possible driving factors. This subject is appropriate for the journal

and the overall presentation is very good. However, there are serious gaps in this work. Specifically, as the authors indicated, to derive group-specific primary production, it requires 1) Chla, 2) aph, 3) PP, and 4) PAR. Because of lacking or no concurrent measurements of these properties, the authors used products in the Kuroshio region generated by NASA or a not commonly used algorithm for aph510 (see Section 2.1.2)! Note that the algorithms for Chla and PP were designed for global mean states, which very likely don't work for this specific (and unique) region (see the wide scatters for the Chla and PP algorithms). Thus, it is critical to evaluate these products derived from satellite for a specific region! But the authors spent no time to do so or to inform the validity of these products in the Kuroshio region. Without such, the group-specific primary production derived from these products are highly questionable to be the least.

In addition, Eq. 1, which plays a key role in this study, is unclear in many ways: 1) Not clear if it is for water-column integrated PP or PP at a specific depth. If it is water-column integrated PP, it is then not proportional to E (e.g., VGPM); if it is PP at a depth, then phi, E, and aph should be in general a function of depth. 2) After the over-simplification (and not proven), the right side of the equation is a spectrum (wavelength dependent), but the left side is a scalar. The two sides are not consistent!

In view of the above, I don't think this work is publishablefifl at least in its current form.

---

## Referee Comment (RC2) · Anonymous Referee #2 · 18 Jul 2017

This manuscript "Seasonal effects of photophysiology and chlorophyll a abundance on phytoplankton group-specific primary production in the Kuroshio region as revealed by SeaStar/SeaWiFS" proposed a new approach to estimate group-specific primary production, along with group-specific quantum yield and chlorophyll a specific absorption, by integrating satellite image processing/statistics with bio-optical models. The method is for the first time developed, yet looks not so solid, partially due to lacking of evaluation of the results. Meanwhile, it is not clear why it is limited to Kuroshio region,

and if the main point is the seasonal effects of photophysiology and chlorophyll a abundance on phytoplankton group-specific primary production, or it is the method, and why that seasonal effects are of concern. In a word, more efforts are needed to make the method more convincing, and to clarify the logic in the manuscript.

Some specific comments: 1. To my knowledge, the target region is actually not only Kuroshio area (see Fig. 1, and the satellite images). Some of the sampling stations are located in the Oyashio region. Almost all diatoms revealed by SeaWiFS are also located in that water (Oyashio). It would be a better choice to focus on Kuroshio-Oyashio frontal region, where water is dynamic and rich of nutrients and various groups of phytoplankton. 2. L28-30: I would suggest to cite references for this sentence "Despite the oligotrophy of the current, higher fishery production has been recognized in and around the current, which is a paradox of the Kuroshio ecosystems." Or data should be provided, proving that higher fishery production was found in the Kuroshio Current compared to other warm currents such as the Gulf Stream. 3. L107-110: Why used CbPM PP, not VGPM or others? Did you do a comparison? 4. Because the method to derive group-specific PP, quantum yield and absorption coefficient is new, a separated section providing details of the method and its evaluation, would be of much more help. 5. Section 3.1, how to understand higher diatom than haptophyte Chl (0.21 vs 0.11) is consistent with higher haptophyte than diatom absorption (0.0026 vs 0.0016), and the notion of more abundant haptophytes in this region? Also, I would say a mean value is not meaningful for this kind of water. 6. In addition, the authors appear careless while preparing this manuscript. For example, Line 404-405 is an incomplete sentence.

---

## Author Comment (AC1) · 8 Aug 2017

Reviewer #1:

AC: We would like to thank the Reviewer for his/her useful comments.

RC: Because different phytoplankton has different response to light and nutrients, it is always desired to know group (or class or species) specific photosynthesis rate,

as such information can also improve the estimation of global primary production. To meet this desire and demand, the authors developed specific primary production in the Kuroshio region for three "functional groups" of phytoplankton: diatoms, haptophytes, and cyanobacteria. Subsequently, the authors described their spatial/temporal variations as well as the possible driving factors. This subject is appropriate for the journal and the overall presentation is very good. However, there are serious gaps in this work. Specifically, as the authors indicated, to derive group-specific primary production, it requires 1) Chla, 2) aph, 3) PP, and 4) PAR. Because of lacking or no concurrent measurements of these properties, the authors used products in the Kuroshio region generated by NASA or a not commonly used algorithm for aph510 (see Section 2.1.2)! Note that the algorithms for Chla and PP were designed for global mean states, which very likely don't work for this specific (and unique) region (see the wide scatters for the Chla and PP algorithms). Thus, it is critical to evaluate these products derived from satellite for a specific region! But the authors spent no time to do so or to inform the validity of these products in the Kuroshio region. Without such, the group-specific primary production derived from these products are highly questionable to be the least.

AC: We should refer to the following paper published earlier, in which validity of the SeaWiFS Chla was evaluated for different regions including the Kuroshio region:

Gregg, W. W. and N. W. Casey: Global and regional evaluation of the SeaWiFS chlorophyll data set, Remote Sens. Environ., 93, 463–479, 2004.

In the above paper, the regional assessment for the North Central Pacific, where a significant number of in situ measurements used for the assessment comes from Japanese waters including the Kuroshio region (see their Fig.1), showed that RMSE % log error of the SeaWiFS Chla is 24.7 (their Fig. 4) with r2=0.71 for N=839 (their Fig. 6). In spite that the SeaWiFS Chla algorithm was developed for the global oceans, the validity of SeaWiFS Chla product was thus shown for the regions that include our study area (but except coastal/shelf waters which were not analyzed in our study). We have cited the above paper in our revised manuscript to support the validity of the global

[Figure]

Chla product in this study. The original satellite primary production (PP) data derived from VGPM (Behenfeld and Falkowski, 1997) was evaluated by Ishizaka et al. (2007) for a Japanese bay within the Kuroshio region. They showed a deviation of VGPM from in situ measurements obtained in the bay, and attributed the deviation to a parameterization issue of Pb_opt (the maximum daily net primary production found in a given water column) used in the VGPM. Therefore, we did not use the original VGPM for our regional study to avoid the known issue, but used CbPM (Westberry et al., 2008) instead, which is more mechanistic in terms of phytoplankton physiology in the sense that the chlorophyll-to-carbon ratio (which can vary spatially) was explicitly taken into account. In the revised manuscript, we have discussed potential errors resulting from our input data.

RC: In addition, Eq. 1, which plays a key role in this study, is unclear in many ways: 1) Not clear if it is for water-column integrated PP or PP at a specific depth. If it is water-column integrated PP, it is then not proportional to E (e.g., VGPM); if it is PP at a depth, then phi, E, and aph should be in general a function of depth. 2) After the oversimplification (and not proven), the right side of the equation is a spectrum (wavelength dependent), but the left side is a scalar. The two sides are not consistent!

AC: We have also clarified the depth-dependency in Eq. 1 in the revised manuscript, such that the last line of the Eq. 1 is:

$$PP(z) = 300 * Phi(z) * PAR(z) * X(lambda,z) * aph(lambda,z)$$

where PAR (or "E" in the reviewer's notation), Phi, X and aph are functions of depth. The first line of Eq. 1 in the original manuscript has been proven useful and well used in relevant scientific communities thus far (e.g. Emerson et al., 1958; Falkowski and Raven, 2007). On the other hand, we would like to stress that our final Eq. 1 (i.e. the last line of Eq. 1) is strictly the same mathematically as the first line. That is, there was no assumption or approximation used to derive the last line of Eq. 1 (hence, there is no over-simplification). As all lines in Eq. 1 in the original manuscript are exactly the

same, we believe that the last line of Eq. 1 should be valid, as long as the first line of Eq. 1 holds.

The notation of the absorption coefficient of phytoplankton appearing in Eq. 1, aph(lambda) only means that aph is a function of wavelength, lambda. For example, aph(lambda) represents a value (scalar) of aph at a certain wavelength, lambda. Thus, it does not represent a vector as a spectrum. Hence, the right- and left-hand sides of the Eq. 1 are consistent. In our manuscript, we distinguished vectors/matrices from scalars by using bold font for the formers (please see Eq. 3 for example). We believe that this writing style is a standard practice in many scientific journals, but we are happy to change the style in a further revised manuscript if it is required by the journal publisher.

References:

Behenfeld, M. J. and Falkowski, P. G.: Photosynthetic rates derived from satellite-based chlorophyll concentration, Limonol. Oceanogr., 42, 1-20, 1997b.

Emerson, R.: The quantum yield of photosynthesis, Annu. Rev. Plant. Physiol., 9, 1–24, 1958.

Falkowski, P. G. and Raven, J. A.: Aquatic Photosynthesis: 2nd Edition, Princeton University Press, New Jersey, USA, 2007.

Ishizaka, J., Siswanto, E., Itoh, T., Murakami, H., Yamaguchi, Y., Horimoto, N., Ishimaru, T., Hashimoto, S., and Saino, T.: Verification of Vertically Generalized Production Model and estimation of primary production in Sagami Bay, Japan, J. Oceanogr., 63, 517-524, 2007.

Westberry, T., Behrenfeld, M. J., Siegel, D. A., and Boss, E.: Carbon-based primary productivity modeling with vertically resolved photoacclimation, Glob. Biogeochem. Cycles, 22, GB2024, doi:10.1029/2007GB003078, 2008.

---

## Author Comment (AC2) · 8 Aug 2017

AC: We would like to thank the Reviewer for his/her useful comments.

RC: This manuscript "Seasonal effects of photophysiology and chlorophyll a abundance on phytoplankton group-specific primary production in the Kuroshio region as revealed by SeaStar/SeaWiFS" proposed a new approach to estimate group-specific primary production, along with group-specific quantum yield and chlorophyll a specific

absorption, by integrating satellite image processing/statistics with bio-optical models. The method is for the first time developed, yet looks not so solid, partially due to lacking of evaluation of the results. Meanwhile, it is not clear why it is limited to Kuroshio region, and if the main point is the seasonal effects of photophysiology and chlorophyll a abun- dance on phytoplankton group-specific primary production, or it is the method, and why that seasonal effects are of concern. In a word, more efforts are needed to make the method more convincing, and to clarify the logic in the manuscript.

AC: To our best knowledge, in situ measurement is generally infeasible (otherwise it has a number of challenges or assumptions) for the group (class level)-specific (i) primary production PP (ii) the quantum yield of photosynthesis and (iii) the optical absorption coefficient aph. Only laboratory experiments using isolated cultures can assess these parameters (e.g. (i) Dikman et al., 2009, (ii) Bidigare et al., 1989; Nielsen, 2008; Morel et al., 1987, (iii) Fujiki and Taguchi, 2002). Thus, it is a shame that a direct comparison between our satellite estimation and in situ measurements is currently not possible. Within this circumstance, we compared our results with literature values for the size-fractionated (yet, not necessarily the group-specific) PP, quantum yield of photosynthesis and aph in our manuscript as our best effort (see L335-417), since the size-fractionated quantities are measureable in situ. Naturally, we found a numerical difference between them, which is expected because of (1) a difference between size-fractionated and taxonomic groups, (2) a difference between an exact timing of size observation in literature and our satellite data, and (3) our methodological uncertainties. However, we also found qualitative consistency between our results and those from literatures. For example, the chla-specific optical absorption was largest for cyanobacteria and smallest for diatoms, diatom/haptophyte PP was larger than cyanobacteria PP. We believe that such comparison effort is still useful, and the agreements found can support our methodology at least qualitatively. Our one of the main points (conclusion) of this manuscript is that the group-specific PP in the Kuroshio region would be regulated by different driving mechanisms, depending on taxonomic groups. We could not derive this conclusion without developing our novel and unique approach (because

no in-situ data is currently available as mentioned earlier). This is the reason why we explained the methodology a bit extensively in our manuscript. In addition, there was a novel finding in developing the methodology, too (i.e. the absorption coefficient at 510nm can be approximated to its spectral average in many cases), therefore the methodology part in our manuscript is also important. We focus on the seasonal cycle as phytoplankton is widely known to have a prominent seasonal cycle. As described in the Introduction (L27-L32), we are interested in the Kuroshio region because of higher fishery production even in the Kuroshio upstream, despite of the fact that the strong Kuroshio western boundary current originates from the oligotrophic waters. This led to the government-funded project which allowed us to conduct our research described here.

RC: 1. To my knowledge, the target region is actually not only Kuroshio area (see Fig. 1, and the satellite images). Some of the sampling stations are located in the Oyashio region. Almost all diatoms revealed by SeaWiFS are also located in that water (Oyashio). It would be a better choice to focus on Kuroshio- Oyashio frontal region, where water is dynamic and rich of nutrients and various groups of phytoplankton.

AC: We agree that the Kuroshio-extension region is interesting in many different scientific aspects (e.g. meso- and smaller scale turbulences along/around the Kuroshio downstream create heterogenic phytoplankton community structure etc.). We consider that those aspects will be better analyzed by other satellite data with higher spatial resolution than SeaWiFS (9 km in resolution) in the near future.

RC: 2. L28-30: I would suggest to cite references for this sentence "Despite the oligotrophy of the current, higher fishery production has been recognized in and around the current, which is a paradox of the Kuroshio ecosystems." Or data should be provided, proving that higher fishery production was found in the Kuroshio Current compared to other warm currents such as the Gulf Stream.

AC: We have cited the following website in the revised manuscript.

http://snf.fra.affrc.go.jp/html/english/index.html

RC: 3. L107-110: Why used CbPM PP, not VGPM or others? Did you do a comparison?

AC: As mentioned above for Reviewer #1, we used CbPM because (1) there is a known issue to use VGPM in our study area (Ishizaka et al., 2007) and (2) CbPM considers variation in the chl-to-carbon ratio more explicitly than VGPM in calculation of PP, therefore physiologically more mechanistic with possibly less implicit assumption(s). We wanted to avoid a dependence of the derivation of our quantum yield index (i.e. the other physiological variable) on the implicit assumptions in VGPM if any. We did not make such a comparison in this study.

RC: 4. Because the method to derive group-specific PP, quantum yield and absorption coefficient is new, a separated section providing details of the method and its evaluation, would be of much more help.

AC: While the derivation of the optical absorption is already sub-sectioned in the original manuscript, we have followed your idea for the description of the group-specific PP and the quantum yield in the revised manuscript.

RC: 5. Section 3.1, how to understand higher diatom than haptophyte Chl (0.21 vs 0.11) is consistent with higher haptophyte than diatom absorption (0.0026 vs 0.0016), and the notion of more abundant haptophytes in this region? Also, I would say a mean value is not meaningful for this kind of water.

AC: Values of chlorophyll-specific absorption coefficient (=aph/Chla) for haptophytes were generally higher than those for diatoms. Therefore, it is possible that haptophyte aph can be higher than diatom aph, even if haptophyte Chla is smaller than diatom Chla. The principle that smaller cells can have the larger Chla-specific absorption coefficient is well-known (Morel and Bricaud, 1981) and the reference has already been cited in our manuscript (L.390-393 in the former manuscript). We suspect that the above led to the absorption coefficient value (aph,i) higher in haptophyte than diatom

in the spatio-temporal average (i.e. climatology). We agree that the average value is less meaningful in interpretation of the value in this particular case. On the other hand, it is also important to describe what magnitude of aph was obtained from our method. As a compromise between these, we have deleted "indicating that haptophytes are more abundant over this region" (L.298) in the revised manuscript in order to avoid a confusion.

RC: In addition, the authors appear careless while preparing this manuscript. For example, Line 404-405 is an incomplete sentence.

AC: Thank you for pointing out the erratum. Below is the complete sentence: However, a care must be taken when smaller spatio-temporal scales (including coastal areas, eddies, local sampling points) are considered, because "spiky" phenomena at higher patio-temporal frequencies (in power spectrum-wise) can mask larger scale variability, so that the above discussions may not necessarily apply to the smaller scale analysis.

References

Bidigare, R. R., Schofield, O., and Prezelin, O.: Influence of zeaxanthin on quantum yield of photosynthesis of Synechococcus clone WH7803 (DC2), Mar. Ecol. Progr. Ser., 56, 177–188, 1989.

Dijkman, N. A., Boschker, H. T. S., Middelburg, J. J., and Kromkamp, J. C.: Group-specific primary production based on stable isotope labaling of phospholipid-derived fatty acids, Limonol. Oceanogr. Methods, 7, 612-625, 2009.

Fujiki, T. and Taguchi, S: Variability in chlorophyll a specific absorption coefficient in marine phytoplankton as a function of cell size and irradiance, J. Plankton. Res., 24, 859–874, 2002.

Ishizaka, J., Siswanto, E., Itoh, T., Murakami, H., Yamaguchi, Y., Horimoto, N., Ishimaru, T., Hashimoto, S., and Saino, T.: Verification of vertically generalized production model and estimation of primary production in Sagami Bay, Japan, J. Oceanogr., 63,

517–524, 2007.

Morel, A., Lazzara, L., and Gostan, J: Growth rate and quantum yield time response for a diatom to changing irradiance (energy and color), Limonl. Oceanogr., 32, 1066–1084, 1987.

Nielsen, M. V.: Photosynthetic characteristics of the cocolithoporid Emiliania huxleyi (Prymnesiophyceae) exposed to elevated concentration of dissolved inorganic carbon, J. Phycol., 31, 715–719, 2008.